# Optimal Sample Complexity for Average Reward Markov Decision Processes

Shengbo Wang, Jose Blanchet & Peter Glynn

Department of Management Science and Engineering
Stanford University
Stanford, CA 94305, USA
{shengbo.wang,jose.blanchet,glynn}@stanford.edu

## Abstract

We resolve the open question regarding the sample complexity of policy learning for maximizing the long-run average reward associated with a uniformly ergodic Markov decision process (MDP), assuming a generative model. In this context, the existing literature provides a sample complexity upper bound of $\widetilde{O}(|S||A|t_{\mathrm{mix}}^2\epsilon^{-2})^*$ and a lower bound of $\Omega(|S||A|t_{\mathrm{mix}}\epsilon^{-2})$. In these expressions, $|S|$ and $|A|$ denote the cardinalities of the state and action spaces respectively, $t_{\mathrm{mix}}$ serves as a uniform upper limit for the total variation mixing times, and $\epsilon$ signifies the error tolerance. Therefore, a notable gap of $t_{\mathrm{mix}}$ still remains to be bridged. Our primary contribution is the development of an estimator for the optimal policy of average reward MDPs with a sample complexity of $\widetilde{O}(|S||A|t_{\mathrm{mix}}\epsilon^{-2})$. This marks the first algorithm and analysis to reach the literature's lower bound. Our new algorithm draws inspiration from ideas in Li et al. (2020), Jin & Sidford (2021), and Wang et al. (2023). Additionally, we conduct numerical experiments to validate our theoretical findings.

## 1 Introduction

This paper offers a theoretical contribution to the area of reinforcement learning (RL) by providing the first provably optimal sample complexity guarantee for a tabular RL environment in which a controller wishes to maximize the *long run average reward* governed by a Markov decision process (MDP).

The landscape of RL has been illuminated by remarkable empirical achievements across a diverse spectrum of applications (Kober et al., 2013; Sadeghi & Levine, 2016; Deng et al., 2017). As a consequence, a great deal of research effort has been channeled into RL theory and its applications within operations research and the management sciences. In many real-world scenarios, influenced by engineering and managerial considerations, the MDP model naturally unfolds within an environment where viable policies must be stable (Bramson, 2008). In these settings, the controlled Markov chain induced by a reasonable policy will typically converge in distribution to a unique steady state, regardless of the initial condition. This phenomenon is known as *mixing*. Within such modeling environments, the long run average reward emerges as a well-defined and pertinent performance measure to maximize. Its relevance is particularly pronounced in scenarios where there is an absence of an inherent time horizon or discount factor. In situations where a system exhibits fast mixing, a finite-time observation of the state process can offer a good statistical representation of its long-term behavior. As a result, it becomes reasonable to anticipate that, for such systems, a lower-complexity algorithm for policy learning is attainable.

Recognized as a significant and challenging open problem in RL theory, the optimal sample complexity for average reward MDPs (AMDPs) under a generative model, i.e., a sampler capable of generating new states for the controlled Markov chain conditioned on any state and action, has been

---

*The $\widetilde{O}, \widetilde{\Omega}, \widetilde{\Theta}$ hide log factors.

Table 1: Sample complexities of AMDP algorithms. When $t_{\text{mix}}$ appear in the sample complexity, an assumption of uniform ergodicity is being made, while the presence of $H^{\ddagger}$ is associated with an assumption that the MDP is weakly communicating.

| Algorithm | Origin | Sample complexity upper bound ($\widetilde{O}$) |
|---|---|---|
| Primal-dual $\pi$ learning | Wang (2017) | $\lvert S \rvert \lvert A \rvert \tau^2 t_{\text{mix}}^2 \epsilon^{-2}$ [†] |
| Primal-dual SMD* | Jin & Sidford (2020) | $\lvert S \rvert \lvert A \rvert t_{\text{mix}}^2 \epsilon^{-2}$ |
| Reduction to DMDP* | Jin & Sidford (2021) | $\lvert S \rvert \lvert A \rvert t_{\text{mix}} \epsilon^{-3}$ |
| Reduction to DMDP | Wang et al. (2022) | $\lvert S \rvert \lvert A \rvert H \epsilon^{-3}$ |
| Refined Q-learning | Zhang & Xie (2023) | $\lvert S \rvert \lvert A \rvert H^2 \epsilon^{-2}$ |
| Reduction to DMDP | This paper | $\lvert S \rvert \lvert A \rvert t_{\text{mix}} \epsilon^{-2}$ |
| Lower bound | Jin & Sidford (2021) | $\Omega(\lvert S \rvert \lvert A \rvert t_{\text{mix}} \epsilon^{-2})$ |
| | Wang et al. (2022) | $\Omega(\lvert S \rvert \lvert A \rvert H \epsilon^{-2})$ |

extensively investigated in the literature (refer to Table 1). In this paper, our focus is on learning an optimal policy for a *uniformly ergodic* AMDP (Meyn & Tweedie, 2009; Wang et al., 2023), and we attain the first optimal sample complexity bound within this context. Specifically, assuming finite state and action spaces, uniform ergodicity implies that the transition matrix $P_\pi$ induced by any stationary Markov deterministic policy $\pi$ has the property that $P_\pi^m$ converges to a matrix with identical rows in $\ell_1$ distance as $m \to \infty$ at a geometric rate. The time constant of this geometric convergence is known as the *mixing time*. The largest mixing time over all $\pi$ is an important parameter denoted by $t_{\text{mix}}$ (see equation (2.2)). In this setting, we establish the following main result:

**Theorem 0** (Informal). *Assuming that the AMDP is uniformly ergodic, the sample complexity of learning a policy that achieves a long run average reward within $\epsilon \in (0, 1]$ of the optimal value with high probability is*

$$\widetilde{\Theta}\left(\frac{\lvert S \rvert \lvert A \rvert t_{\text{mix}}}{\epsilon^2}\right),$$

*where $\lvert S \rvert, \lvert A \rvert$ are the cardinality of the state and action spaces, respectively.*

A rigorous version of Theorem 0 is presented in Theorem 2. We highlight that this is the first optimal result in the domain of sample complexity of AMDPs, as it achieves the lower bound in Jin & Sidford (2021) for uniformly ergodic AMDPs.

## 1.1 LITERATURE REVIEW

In this section, we review pertinent literature to motivate our methodology and draw comparisons with our contributions.

**Sample Complexity of Average Reward RL:** The relevant literature is summarized in Table 1. It's important to note that the works mentioned in this table revolve around the concept of mixing, albeit from distinct perspectives. On one side, Wang (2017); Jin & Sidford (2020; 2021), and the current paper assumes a uniformly ergodic AMDP. Conversely, Wang et al. (2022) and Zhang & Xie (2023) operate under a milder assumption, considering weakly communicating AMDPs as defined in Puterman (1994). Under this assumption, only the optimal average reward is guaranteed to remain independent of the initial state and is attained by a stationary Markov deterministic policy. In particular, certain policies in this setting might not lead to a mixing Markov chain, potentially rendering the policy's average reward state dependent. Consequently, the uniform mixing time upper bound $t_{\text{mix}}$ is infinite within such instances. In this context, a pertinent complexity metric parameter denoted as $H$ is the *span semi-norm* (see (A.1)) upper bound of transient value functions $H = \max_{\bar{\pi}} \lvert u^{\bar{\pi}} \rvert_{\text{span}}$, where the max is taken over all optimal policies and $u^{\bar{\pi}}$ is defined in (2.5)[‡‡]. As demonstrated in

---

[‡] $H$ is the maximum span of optimal transient value functions. See the literature review for a discussion.

[†] Let $\eta^\pi$ be the stationary distribution of $P_\pi$, then $\tau := \max_{\pi \in \Pi} \max_{s \in S} \eta^\pi(s) / \min_s \eta^\pi(s)$.

[*] SMD stands for stochastic mirror descent while DMDP stands for discounted MDP.

[‡‡] For periodic chains, the limit in (2.5) is understood as the Cesaro limit, see Puterman (1994).

Table 2: Sample complexity of uniformly ergodic DMDP algorithms when $t_{\mathrm{mix}} \le (1-\gamma)^{-1}$, where $\gamma$ is the discount factor.

| Origin | Sample complexity upper bound ($\widetilde{O}$) | Minimum sample size ($\widetilde{\Omega}$) |
|---|---|---|
| Azar et al. (2013) | $|S||A|(1-\gamma)^{-3}\epsilon^{-2}$ | $|S||A|(1-\gamma)^{-3}$ |
| Agarwal et al. (2020) | $|S||A|(1-\gamma)^{-3}\epsilon^{-2}$ | $|S||A|(1-\gamma)^{-2}$ |
| Li et al. (2020) | $|S||A|(1-\gamma)^{-3}\epsilon^{-2}$ | $|S||A|(1-\gamma)^{-1}$ |
| Wang et al. (2023) | $|S||A|t_{\mathrm{mix}}(1-\gamma)^{-2}\epsilon^{-2}$ | $|S||A|(1-\gamma)^{-3}$ |
| This paper | $|S||A|t_{\mathrm{mix}}(1-\gamma)^{-2}\epsilon^{-2}$ | $|S||A|(1-\gamma)^{-1}$ |
| Lower bound Wang et al. (2023) | $t_{\mathrm{mix}}(1-\gamma)^{-2}\epsilon^{-2}$ [§] | N/A |

Wang et al. (2022), it holds that $H \le 8t_{\mathrm{mix}}$ when the reward is bounded in $[0,1]$. It's important to note that $H$ depends on the specific reward function, whereas $t_{\mathrm{mix}}$ does not. Therefore, the reverse inequality cannot hold, even within the realm of uniformly ergodic MDPs. This is evident when we consider a reward function that is identically 0. However, it's worth mentioning that the family of worst-case uniformly ergodic MDP instances presented in Wang et al. (2023) exhibits $H = \Theta(t_{\mathrm{mix}})$. See Section 5 for some more discussion.

Additionally, we highlight that two existing papers in the literature, namely Jin & Sidford (2021) and Wang et al. (2022), employ the reduction to a discounted MDP (DMDP) method to facilitate policy learning for AMDPs. This paper follows the same approach. In Section 1.2, we will offer an in-depth discussion of this methodology and a comparison with these two papers.

**Sample Complexity of Discounted Tabular RL:** In recent years, there has been substantial interest in understanding the worst-case sample complexity theory of tabular DMDPs. This research has given rise to two primary approaches: model-based and model-free. Model-based strategies involve constructing an empirical MDP model from data and employing dynamic programming (see Azar et al. (2013); Sidford et al. (2018); Agarwal et al. (2020); Li et al. (2022)), yielding optimal upper and lower bounds of $\widetilde{\Theta}(|S||A|(1-\gamma)^{-3}\epsilon^{-2})$, where $\gamma$ is the discount factor. Meanwhile, the model-free route maintains lower-dimensional statistics of the transition data, as exemplified by the iconic Q-learning (Watkins & Dayan, 1992) and its extensions. Li et al. (2021) demonstrates that the classic Q-learning algorithm has a worst-case sample complexity of $\widetilde{\Theta}(|S||A|(1-\gamma)^{-4}\epsilon^{-2})$. However, Wainwright (2019) introduced a variance-reduced Q-learning variant that reaches the same lower bound as model-based methods.

Worst-case analysis provides uniform guarantees for convergence rates across all $\gamma$-discounted MDP instances. However, the worst-case examples that achieve the lower bound must have a transition kernel or reward function that depends on $\gamma$ (Wang et al., 2023). In contrast, Khamaru et al. (2021) focuses on *instance-dependent* settings, where the transition kernel and reward function are fixed, and proves matching sample complexity upper and lower bounds. Wang et al. (2023) concentrates on scenarios in which the class of MDPs can encompass arbitrary reward functions, while the transition kernels are assumed to obey various mixing time upper bounds. A particular setting is when this mixing time upper bound is uniform across all policies. This specific setting aligns with the central objective of this paper: Derive optimal sample complexity theory for uniformly ergodic AMDPs.

## 1.2 ALGORITHM METHODOLOGY

Our approach to algorithmic design draws inspiration from Jin & Sidford (2021), wherein the optimal policy of a uniformly ergodic AMDP is approximated by that learned from a discounted MDP (DMDP) with a discount factor $1 - \gamma = \Theta(\epsilon/t_{\mathrm{mix}})$. This idea of approximating the AMDP by a DMDP been considered and implemented since 1970s, see Hordijk & Tijms (1975). It is known as the *reduction* method in the AMDP sample complexity literature. As shown in Table 1, however,

---

[§]The lower bound in Wang et al. (2023) assumes $|S||A| = O(1)$.

both Jin & Sidford (2021) and Wang et al. (2022) employed this strategy and yet obtained an $\epsilon^{-3}$ dependency, deviating from the canonical $\epsilon^{-2}$ rate obtained in this paper.

To understand this deviation and motivate our methodology, we provide a brief discussion of the sample complexity theory for uniformly ergodic DMDPs. Prior to Wang et al. (2023), the DMDP algorithm, known as perturbed model-based planning, and the analysis in Li et al. (2020) achieved a sample complexity dependence on $1 - \gamma$ of $\widetilde{O}((1 - \gamma)^{-3})$. Though optimal for the worst-case DMDP, this dependence on $1 - \gamma$ is sub-optimal for policy learning of uniformly ergodic DMDPs. On the other hand, the state-of-the-art algorithm and associated analysis in Wang et al. (2023) yield an optimal $\widetilde{\Theta}(t_{\mathrm{mix}}(1-\gamma)^{-2})$ dependence, as displayed in Table 2. The sub-optimal $\epsilon^{-3}$ dependency observed in both Jin & Sidford (2021) and Wang et al. (2022) can be attributed to their utilization of the $\widetilde{O}((1 - \gamma)^{-3})$ result from Li et al. (2020), without taking into account the complexity reduction from the associated mixing assumptions.

Building upon the aforementioned DMDP results, our strategy is rooted in recognizing that the optimal sample complexity of uniformly ergodic DMDPs is $\widetilde{\Theta}(|S||A|t_{\mathrm{mix}}(1-\gamma)^{-2}\epsilon^{-2})$. As shown in Table 2, the algorithm presented in Wang et al. (2023) is capable of achieving this complexity. However, as indicated in the third column of Table 2, it necessitates a minimum sample size of $\widetilde{\Omega}(|S||A|(1-\gamma)^{-3})$. This quantity can be interpreted as the initial setup cost of the algorithm which is indifferent to the specification of $\epsilon$. Unfortunately, $\widetilde{\Omega}(|S||A|(1-\gamma)^{-3})$ proves to be excessively large for our objective. For a more comprehensive discussion on this issue, see Section 3.1.

To overcome this challenge, in this paper, we successfully establish an optimal sample complexity upper bound for the algorithm proposed in Li et al. (2020) in the setting of uniformly ergodic DMDPs. We achieve this while simultaneously maintaining a minimum sample size requirement of $\widetilde{\Omega}(|S||A|(1-\gamma)^{-1})$. This optimal sample complexity bound for uniformly ergodic DMDPs, which builds upon and enhances the findings in Wang et al. (2023), is of independent theoretical significance. The formal statement is presented in Theorem 1, Section 3.1. This accomplishment is made possible by applying analytical techniques presented in Wang et al. (2023). In conjunction with the reduction methodology outlined in Jin & Sidford (2021), these developments collectively result in the AMDP Algorithm 2, which attains the optimal sample complexity as outlined in Theorem 0.

## 2 MARKOV DECISION PROCESSES: DEFINITIONS

We consider the setting of MDPs with finite state and action space $S$ and $A$. Let $\mathcal{P}(S)$ denote the set of probability measures on $S$. An element $p \in \mathcal{P}(S)$ can be seen as a row vector in $\mathbb{R}^{|S|}$. Let $\mathcal{M}(r, P, \gamma)$ denote a discounted MDP (DMDP), where $r : S \times A \to [0, 1]$ is the reward function, $P : S \times A \to \mathcal{P}(S)$ is the transition kernel, and $\gamma \in (0, 1)$ is the discount factor. Note that $P$ can be identified with a $s, a$ indexed collection of measures $\{p_{s,a} \in \mathcal{P}(S) : (s, a) \in S \times A\}$. We denote an average reward MDP (AMDP) with the same reward function and transition kernel by $\bar{\mathcal{M}}(r, P)$.

Let $\mathbf{H} = (|S| \times |A|)^{\mathbb{Z}_{\geq 0}}$ and $\mathcal{H}$ the product $\sigma$-field form the underlying measureable space. Define the stochastic process $\{(X_t, A_t), t \geq 0\}$ by the point evaluation $X_t(h) = s_t, A_t(h) = a_t$ for all $t \geq 0$ for any $h = (s_0, a_0, s_1, a_1, \dots) \in \mathbf{H}$. At each time and current state $X_t$, if action $A_t$ is chosen, the decision maker will receive a reward $r(X_t, A_t)$. Then, the law of the subsequent state satisfies $\mathcal{L}(X_{t+1}|X_0, A_0, \dots, X_t, A_t) = p_{X_t, A_t}(\cdot)$ w.p.1.

It is well known that to achieve optimal decision making in the context of infinite horizon AMDPs or DMDPs (to be introduced), it suffices to consider the policy class $\Pi$ consisting of stationary, Markov, and deterministic policies; i.e. any $\pi \in \Pi$ can be seen as a function $\pi : S \to A$. For $\pi \in \Pi$ and initial distribution $\mu \in \mathcal{P}(S)$, there is a unique probability measure $Q_\mu^\pi$ on the product space s.t. the chain $\{(X_t, A_t), t \geq 0\}$ has finite dimensional distributions

$$Q_\mu^\pi(X_0 = s_0, A_0 = a_0, \dots, A_t = a_t) = \mu(s_0)p_{s_0, \pi(s_0)}(s_1) \dots p_{s_{t-1}, \pi(s_{t-1})}(s_t)\mathbb{1}\left\{\pi(s_i) = a_i, \forall i\right\},$$

where $\mathbb{1}$ is the indicator function. Note that this also implies that $\{X_t, t \geq 0\}$ is a Markov chain under $Q_\mu^\pi$ with transition matrix $P_\pi$ defined by

$$P_\pi(s, s') = p_{s, \pi(s)}(s').$$

Also, we define $r_\pi$ by $r_\pi(s) = r(s, \pi(s))$.

## 2.1 DISCOUNTED MDPS

For $\pi \in \Pi$, let $E_\mu^\pi$ denote the expectation under under $Q_\mu^\pi$. For $\mu$ with full support $S$, the *discounted value function* $v^\pi(s)$ of a DMDP is defined via

$$v^\pi(s) := E_\mu^\pi \left[ \sum_{t=0}^\infty \gamma^t r(X_t, A_t) \middle| X_0 = s \right].$$

It can be seen as a vector $v^\pi \in \mathbb{R}^{|S|}$, and can be computed using the formula $v^\pi = (I - \gamma P_\pi)^{-1} r_\pi$. The optimal discounted value function is defined as

$$v^*(s) := \max_{\pi \in \Pi} v^\pi(s), \quad s \in S.$$

For probability measure $p$ on $S$, let $p[v]$ denote the sum $\sum_{s \in S} p(s)v(s)$. It is well known that $v^*$ is the unique solution of the following *Bellman equation*:

$$v^*(s) = \max_{a \in A} \left( r(s, a) + \gamma p_{s,a}[v^*] \right). \tag{2.1}$$

Moreover, $\pi^*(s) \in \arg\max_{a \in A} \left( r(s, a) + \gamma p_{s,a}[v^*] \right)$ is optimal and hence $v^* = (I - \gamma P_{\pi^*})^{-1} r_{\pi^*}$.

## 2.2 AVERAGE REWARD MDP

As explained in the introduction, in this paper, we assume that the MDP of interest is uniformly ergodic. That is, for all $\pi \in \Pi$, $P_\pi$ is uniformly ergodic. By Wang et al. (2023), this is equivalent to the setting in (Wang, 2017; Jin & Sidford, 2021) where the authors assume

$$t_{\text{mix}} := \max_{\pi \in \Pi} \inf \left\{ m \geq 1 : \max_{s \in S} \| P_\pi^m(s, \cdot) - \eta_\pi(\cdot) \|_1 \leq \frac{1}{2} \right\} < \infty. \tag{2.2}$$

Here $\eta_\pi(\cdot)$ is the stationary distribution of $P_\pi$ and $\|\cdot\|_1$ is $\ell_1$ distance between two probability vectors. Further, this is also shown in Wang et al. (2023) to be equivalent to the following assumption:

**Assumption 1** (Uniformly Ergodic MDP). *For any $\pi \in \Pi$, there exists $m \geq 1, q \leq 1$ and probability measure $\psi \in \mathcal{P}(S)$ such that for all $s \in S$, $P_\pi^m(s, \cdot) \geq q\psi(\cdot)$.*

Here, the notation $P_\pi^m$ denotes the $m$th power of the matrix $P_\pi$. Assumption 1 is commonly referred to, in the general state-space Markov chain literature, as $P_\pi$ satisfying the *Doeblin condition* (Meyn & Tweedie, 2009). In the context of Assumption 1, we define the minorization time as follows.

**Definition 1** (Minorization Time). Define the minorization time for a uniformly ergodic kernel $P_\pi$ as

$$t_{\text{minorize}}(P_\pi) := \inf \left\{ m/q : \inf_{s \in S} P_\pi^m(s, \cdot) \geq q\psi(\cdot) \text{ for some } \psi \in \mathcal{P}(S) \right\}.$$

Define the minorization time for the uniformly ergodic MDP as

$$t_{\text{minorize}} := \max_{\pi \in \Pi} t_{\text{minorize}}(P_\pi).$$

Since $\Pi$ is finite, the above $\max$ is always achieved, and hence $t_{\text{minorize}} < \infty$. Moreover, Theorem 1 of Wang et al. (2023) shows that $t_{\text{minorize}}$ and $t_{\text{mix}}$ are equivalent up-to constant factors:

$$t_{\text{minorize}} \leq 22 t_{\text{mix}} \leq 22 \log(16) t_{\text{minorize}}. \tag{2.3}$$

Therefore, the subsequent complexity dependence written in terms of $t_{\text{minorize}}$ can be equivalently expressed using $t_{\text{mix}}$.

Under Assumption 1, for any initial distribution $X_0 \sim \mu$, the *long run average reward* of any policy $\pi \in \Pi$ is defined as

$$\alpha^\pi := \lim_{T \to \infty} \frac{1}{T} E_\mu^\pi \left[ \sum_{t=0}^{T-1} r(X_t, A_t) \right]$$

where the limit always exists and doesn't depend on $\mu$. The long run average reward $\alpha^\pi$ can be characterized via any solution pair $(u, \alpha)$, $u : S \to \mathbb{R}$ and $\alpha \in \mathbb{R}$ to the *Poisson's equation*,

$$r_\pi - \alpha = (I - P_\pi)u. \tag{2.4}$$

Under Assumption 1, a solution pair $(u, \alpha)$ always exists and is unique up to a shift in $u$; i.e. $\{(u + ce, \alpha) : c \in \mathbb{R}\}$, where $e(s) = 1, \forall s \in S$, are all the solution pairs to (2.4). In particular, for any $\pi \in \Pi$, define the *transient value function* a.k.a. the *bias* as

$$u^\pi(s) := \lim_{T \to \infty} E_s^\pi \left[ \sum_{t=0}^{T-1} (r_\pi(X_t) - \alpha^\pi) \right] \tag{2.5}$$

where the limit always exists. Then, $(u^\pi, \alpha^\pi)$ is the unique up to a shift solution to (2.4).

Similar to DMDPs, define the optimal long run average reward $\bar{\alpha}$ as

$$\bar{\alpha} := \max_{\pi \in \Pi} \alpha^\pi.$$

Then, for any $\bar{\pi}$ that achieve the above maximum, $(u^{\bar{\pi}}, \bar{\alpha})$ solves $r_{\bar{\pi}} - \bar{\alpha} = (I - P_{\bar{\pi}}) u^{\bar{\pi}}$.

## 3 OPTIMAL SAMPLE COMPLEXITIES UNDER A GENERATIVE MODEL

In this section, we aim to develop an algorithm for AMDPs that achieves the sample complexity as presented in Theorem 0. The randomness used by the algorithm arises from an underlying probability space $(\Omega, \mathcal{F}, P)$. We proceed under the assumption that we have access to a *generative model*, or sampler, which allows us to independently draw samples of the subsequent state from the transition probability $\{p_{s,a}(s') : s' \in S\}$ given any state action pair $(s, a)$.

As explained in Section 1.2, our methodology closely aligns with the approach introduced in Jin & Sidford (2021). We leverage the policy acquired from a suitably defined DMDP as an approximation to the optimal policy for the AMDP. However, prior to this work, the state-of-the-art DMDP algorithms, along with the accompanying worst-case sample complexity analysis, fall short in achieving the optimal sample complexity articulated in Theorem 0 (Jin & Sidford, 2021; Wang et al., 2022). This limitation emanates from the fact that the sample complexity of uniformly ergodic DMDPs has a dependence $(1 - \gamma)^{-2}$, as demonstrated in the preceding research Wang et al. (2023). This is a notable improvement over the $(1 - \gamma)^{-3}$ dependence associated with the worst-case-optimal DMDP theory (Li et al., 2020), a foundation upon which Jin & Sidford (2021) and Wang et al. (2022) were constructed. Consequently, to attain the desired complexity result, our initial step involves establishing enhanced sample complexity assurances for uniformly ergodic DMDPs.

### 3.1 A SAMPLE EFFICIENT ALGORITHM FOR UNIFORMLY ERGODIC DMDPS

We recognize that the optimal sample complexity of uniformly ergodic DMDPs should be $\widetilde{\Theta}(|S||A|t_{\mathrm{minorize}}(1 - \gamma)^{-2}\epsilon^{-2})$; c.f. (Wang et al., 2023). Regrettably, as mentioned earlier, the algorithm introduced in Wang et al. (2023), while indeed capable of achieving this sample complexity, falls short in terms of the minimum sample size as it requires $n = \widetilde{\Omega}(|S||A|(1 - \gamma)^{-3})$ for its execution, a complexity that is too large for our purposes.

To elaborate on this issue, we note that the algorithm introduced in Wang et al. (2023) constitutes a variant of the variance-reduced Q-learning (Wainwright, 2019). This family of algorithms necessitates a minimum sample size of $n = \widetilde{\Omega}(|S||A|(1 - \gamma)^{-3})$, as per existing knowledge. Remarkably, model-based algorithms can achieve significantly smaller minimal sample sizes: e.g. $n = \widetilde{\Omega}(|S||A|(1 - \gamma)^{-2})$ in Agarwal et al. (2020), and the full-coverage $n = \widetilde{\Omega}(|S||A|(1 - \gamma)^{-1})$ in Li et al. (2020). See Table 2. This comparison motivates our adoption of the algorithm introduced in Li et al. (2020) and to draw upon the techniques utilized for analyzing mixing MDPs from Wang et al. (2023). The synergistic combination of these ideas, as elucidated in Theorem 1, results in an algorithm with the optimal sample complexity of $\widetilde{\Theta}(|S||A|t_{\mathrm{minorize}}(1 - \gamma)^{-2}\epsilon^{-2})$ and minimum sample size $n = \widetilde{\Omega}(|S||A|(1 - \gamma)^{-1})$ at the same time.

In this section, our analysis suppose Assumption 1, $t_{\mathrm{minorize}} \leq (1 - \gamma)^{-1}$, and $\gamma \geq \frac{1}{2}$.

### 3.1.1 THE DMDP ALGORITHM AND ITS SAMPLE COMPLEXITY

We introduce the perturbed model-based planning algorithm for DMDPs in Li et al. (2020). Let $\zeta > 0$ be a design parameter that we will specify later. Consider a random perturbation of the

reward function

$$R(s,a) := r(s,a) + Z(s,a), \quad (s,a) \in S \times A$$

where the random element $Z : S \times A \to [0, \zeta]$

$$Z(s,a) \sim \mathrm{Unif}(0, \zeta) \tag{3.1}$$

i.i.d. $\forall (s,a) \in S \times A$. The reason to consider a perturbed reward with amplitude parameter $\zeta$ is to ensure that the optimality gap of the optimal policy, compared with any other suboptimal policy, is sufficiently large. To accomplish this, it is not necessary to assume uniform distributions of $Z(s,a)$. However, we opt for this choice for the sake of convenience. This reward perturbation technique is well motivated in Li et al. (2020). So, we refer the readers to this paper for a detailed exposition. Then, the perturbed model-based planning algorithm therein is formulated in Algorithm 1.

---

**Algorithm 1** Perturbed Model-based Planning (Li et al., 2020): PMBP$(\gamma, \zeta, n)$

---

**Input:** Discount factor $\gamma \in (0,1)$. Perturbation amplitude $\zeta > 0$. Sample size $n \geq 1$.
Sample $Z$ as in (3.1) and compute $R = r + Z$.
Sample i.i.d. $S_{s,a}^{(1)}, S_{s,a}^{(2)}, \ldots, S_{s,a}^{(n)}$, for each $(s,a) \in S \times A$. Then, compute the *empirical kernel* $\widehat{P} := \{\hat{p}_{s,a}(s') : (s,a) \in S \times A, s' \in S\}$ where

$$\hat{p}_{s,a}(s') := \frac{1}{n} \sum_{i=1}^{n} \mathbb{1}\left\{S_{s,a}^{(i)} = s'\right\}, \quad (s,a) \in S \times A.$$

Compute the solution $\hat{v}_0$ to the empirical version of the Bellman equation (2.1); i.e. $\forall s \in S$, $\hat{v}_0(s) = \max_{a \in A} \left(R(s,a) + \gamma \hat{p}_{s,a}[\hat{v}_0]\right)$. Then, extract the greedy policy

$$\hat{\pi}_0(s) \in \arg\max_{a \in A} \left(R(s,a) + \gamma \hat{p}_{s,a}[\hat{v}_0]\right), \quad s \in S.$$

**return** $\hat{\pi}_0$.

---

By (2.1), $\hat{\pi}_0$ returned by Algorithm 1 optimal for the DMDP instance $\mathcal{M}(R, \widehat{P}, \gamma)$. Also, computing $\hat{v}_0$ therein involves solving a fixed point equation in $\mathbb{R}^{|S|}$. This can be done (with machine precision) by performing value or policy iteration using no additional samples.

In summary, Li et al. (2020) proposed to use $\hat{\pi}_0$, the optimal policy of the reward-perturbed empirical DMDP $\mathcal{M}(R, \widehat{P}, \gamma)$, as the estimator to the optimal policy $\pi^*$ of the DMDP $\mathcal{M}(r, P, \gamma)$. They proved that this procedure achieves a sample complexity upper bound of $\widetilde{\Theta}(|S||A|(1-\gamma)^{-3}\epsilon^{-2})$ over all DMDPs (not necessarily uniformly ergodic ones) with a minimum sample size requirement $n = \widetilde{\Omega}(|S||A|(1-\gamma)^{-1})$. We are able to improve the analysis and achieve an accelerated result of independent interest in the context of uniformly ergodic DMDPs. Before introducing our results, let us define two parameters. Recall $\zeta$ from (3.1). For prescribed error probability $\delta \in (0,1)$, define

$$\beta_\delta(\eta) = 2\log\left(\frac{24|S||A|\log_2((1-\gamma)^{-1})}{(1-\gamma)^2\eta\delta}\right) \text{ and } \eta_\delta^* = \frac{\zeta\delta(1-\gamma)}{9|S||A|^2}.$$

The reason for defining $\beta_\delta(\cdot)$ and $\eta_\delta^*$ will become clear when we introduce the intermediate results in Proposition A.1 and A.2 in the Appendix. Now, we are ready to state improved error and sample complexity bounds for Algorithm 1 that achieve minmax optimality for uniformly ergodic DMDPs:

**Theorem 1.** *Suppose Assumption 1 is in force. Then, for any $\gamma \in [1/2, 1)$, $\zeta > 0$, and $n \geq 64\beta_\delta(\eta_\delta^*)(1-\gamma)^{-1}$, the policy $\hat{\pi}_0$ returned by Algorithm 1 PMBP$(\gamma, \zeta, n)$ satisfies*

$$0 \leq v^* - v^{\hat{\pi}_0} \leq \frac{2\zeta}{1-\gamma} + 486\sqrt{\frac{\beta_\delta(\eta_\delta^*)t_{\mathrm{minorize}}}{(1-\gamma)^2 n}} \tag{3.2}$$

*w.p. at least $1-\delta$. Consequently, choose $\zeta = (1-\gamma)\epsilon/4$, the sample complexity to achieve an error $0 < \epsilon \leq \sqrt{t_{\mathrm{minorize}}/(1-\gamma)}$ w.p. at least $1-\delta$ is*

$$\widetilde{O}\left(\frac{|S||A|t_{\mathrm{minorize}}}{(1-\gamma)^2\epsilon^2}\right).$$

The proof of Theorem 1 is deferred to Appendix A.

*Remark.* Compare to the worst case result in, e.g., Azar et al. (2013) and Li et al. (2020), Theorem 1 replaces a power of $(1 - \gamma)^{-1}$ by $t_{\text{minorize}}$ in (3.2). The implied sample complexity upper bound matches the lower bound in Wang et al. (2023) up to log factors. So, $\widetilde{O}$ can be replaced by $\widetilde{\Theta}$. Also, this is achieved with a minimum sample size $n = 64\beta_\delta(\eta_\delta^*)(1 - \gamma)^{-1}$. This and the mixing time equivalence (2.3) confirms the sample complexity claim in Table 2.

## 3.2 An Optimal Sample Complexity Upper Bound for AMDPs

Using the sample complexity upper bound in Theorem 1, we then adapt the reduction procedure considered in Jin & Sidford (2021). This leads to an algorithm that learns an $\epsilon$-optimal policy for any AMDP satisfying Assumption 1 with the optimal sample complexity. Concretely, we run the reduction and perturbed model-based planning Algorithm 2.

---

**Algorithm 2** Reduction and Perturbed Model-based Planning

**Input:** Error tolerance $\epsilon \in (0, 1]$.
Assign

$$\gamma = 1 - \frac{\epsilon}{19t_{\text{minorize}}}, \quad \zeta = \frac{1}{4}(1 - \gamma)t_{\text{minorize}}, \quad \text{and } n = \frac{c\beta_\delta(\eta_\delta^*)}{(1 - \gamma)^2 t_{\text{minorize}}}$$

where $c = 4 \cdot 486^2$.
Run Algorithm 1 with parameter specification PMBP$(\gamma, \zeta, n)$ and obtain output $\hat{\pi}_0$.
**return** $\hat{\pi}_0$.

---

This algorithm has the following optimal sample complexity guarantee:

**Theorem 2.** *Suppose Assumption 1 is in force. The policy $\hat{\pi}_0$ output by Algorithm 2 satisfies $0 \leq \bar{\alpha} - \alpha^{\hat{\pi}_0} \leq \epsilon$ w.p. at least $1 - \delta$. Moreover, the total number of samples used is*

$$\widetilde{O}\left(\frac{|S||A|t_{\text{minorize}}}{\epsilon^2}\right).$$

The proof of Theorem 2 is deferred to Appendix B.

*Remark.* This achieves the minmax lower bound in Jin & Sidford (2021) up to log factors. So, $\widetilde{O}$ can be replaced by $\widetilde{\Theta}$. This and the equivalence relationship (2.3) between $t_{\text{minorize}}$ and $t_{\text{mix}}$ is a formal statement of Theorem 0.

## 4 Numerical Experiments

In this section, we conduct two numerical experiments to verify our algorithm's optimal sample complexity dependence on $\epsilon$ and $t_{\text{minorize}}$. The family of reward functions and transition kernels used for both experiments belongs to the family of hard instances constructed in Wang et al. (2023). By the same reduction argument, this family of AMDPs has sample complexity $\Omega(t_{\text{minorize}}\epsilon^{-2})$.

First, we verify the $\epsilon$ dependence of our algorithm. To achieve this, we study the error of estimating the true average reward $\bar{\alpha} = 0.5$ as the sample size increases. The experiment runs 300 replications of Algorithm 2 under different sample sizes and constructs the red data points and regression line in Figure 1a. We also implement the algorithm proposed in Jin & Sidford (2021), conduct the same experiment, and produce the data in blue. Our algorithm outperforms the prior work. Moreover, in log-log scale, the red regression line has a slope that is close to $-1/2$, indicating a $n^{-1/2}$ convergence rate. This verifies the optimal $\widetilde{O}(\epsilon^{-2})$ dependence of our Algorithm 2. On the other hand, the blue line has a slope near $-1/3$, indicating a suboptimal $\widetilde{O}(\epsilon^{-3})$ for the algorithm in Jin & Sidford (2021). This significant improvement is due to our optimal implementation and analysis of the DMDP algorithm in Li et al. (2020), reducing sample complexity dependence on $(1 - \gamma)^{-1}$.

Next, we verify our algorithm's sample complexity dependence on $t_{\text{minorize}}$. Fix a $\epsilon > 0$ and recall that $1 - \gamma = \Theta(\epsilon/t_{\text{minorize}}) = \Theta(t_{\text{minorize}}^{-1})$. So, the optimal sample size for Algorithm

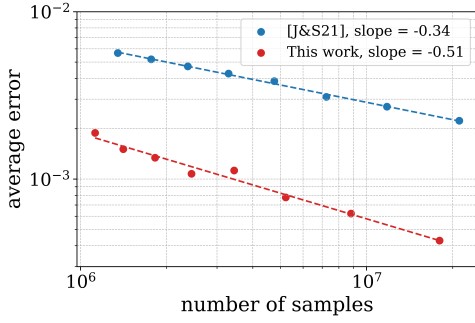 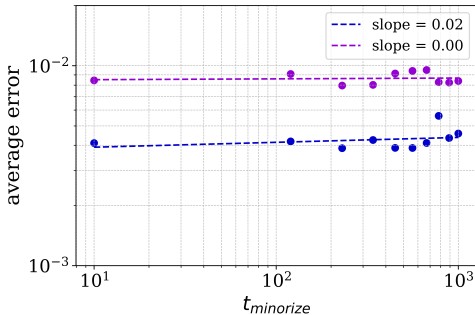

(a) Convergence rate comparison with Jin & Sidford (2021). A $-0.5$ slope verifies the $\widetilde{O}(\epsilon^{-2})$ dependence.

(b) Verification of $t_{\text{minorize}}$ dependence. A 0 slope indicates the $\widetilde{O}(t_{\text{minorize}})$ dependence.

Figure 1: Numerical experiments using the hard MDP instance in Wang et al. (2023).

2 is $n = \widetilde{\Theta}((1-\gamma)^{-2}t_{\text{minorize}}^{-1}) = \widetilde{\Theta}\left(t_{\text{minorize}}\right)$ which is linear in $t_{\text{minorize}}$. Thus, applying Algorithm 2 to the hard instance with minorization time $t_{\text{minorize}}$ using sample size $n = Ct_{\text{minorize}}$ for some large constant $C$, our sample complexity upper bound suggests that the estimation error of $\bar{\alpha}$ should be $\widetilde{O}(1)$. Consequently, plotting the average error against the number of samples used or, equivalently, $t_{\text{minorize}}$ on a log-log scale while varying $t_{\text{minorize}}$ should yield a line with a near 0 (or possibly a negative) slope, according to our theory. The experiments in Figure 1b use $C = 4500$ for the purple line and $C = 18000$ for the blue line. With $t_{\text{minorize}}$ varying within the range $[10, 1000]$, both regression lines exhibit a near 0 slope, indicating a $\widetilde{O}(t_{\text{minorize}})$ dependence.

Therefore, through the numerical experiments presented in Figure 1, we separately verify the optimality of our algorithm's sample complexity dependence on both $\epsilon$ and $t_{\text{minorize}}$.

## 5 CONCLUDING REMARKS

We now discuss certain limitations intrinsic to our proposed methodology as well as potential avenues for future research. Firstly, the use of the DMDP approximation approach necessitates a priori knowledge of an upper bound on the uniform mixing time for the transition kernel $P$ in order to initiate the algorithm. In practical applications, such an upper bound can be challenging to obtain, or in certain instances, result in excessively pessimistic estimates. One could circumvent this by using an additional logarithmic factor of samples, thus allowing for the algorithm to operate with geometrically larger values of $t_{\text{mix}}$, and terminate after a suitable number of iterations. Nevertheless, this approach still hinges upon the knowledge of the mixing time to achieve optimal termination.

Secondly, we assume a strong form of MDP mixing known as uniform ergodicity. While theoretically optimal, this notion of mixing yields conservative upper bounds on sample complexity in situations wherein suboptimal policies induce Markov chains with especially large mixing times. It is our contention, supported by the findings presented in references such as Wang et al. (2022) and Wang et al. (2023), that the sample complexity should be contingent solely upon the properties of the optimal policies. Moreover, the complexity measure parameter $H$ that is used in Wang et al. (2022); Zhang & Xie (2023) has several advantages over $t_{\text{mix}}$ that lie beyond the ability to generalize to weakly communicating MDPs. In particular, for periodic chains and special situations where the optimal average reward remains state-independent despite the presence of multiple recurrent classes of the transition kernel induced by an optimal policy, the parameter $H$ is well-defined but $t_{\text{mix}}$ is not (Puterman, 1994). Consequently, we are now dedicating research effort to the development of an algorithm and analysis capable of achieving a sample complexity of $\widetilde{O}(|S||A|H\epsilon^{-2})$.

Lastly, as the assumption of uniform ergodicity extends beyond finite state space MDPs, we aspire to venture into the realm of general state-space MDPs. Our objective is to extrapolate the principles underpinning our methodology to obtain sample complexity results for general state space MDPs.

ACKNOWLEDGMENTS

The material in this paper is based upon work supported by the Air Force Office of Scientific Research under award number FA9550-20-1-0397. Additional support is gratefully acknowledged from NSF 1915967, 2118199, 2229012, 2312204.

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

# Appendices

## A    STATISTICAL PROPERTIES OF THE ESTIMATORS OF UNIFORMLY ERGODIC DMDPS

In this section, our objective is to establish concentration properties pertaining to the estimators of the value function and optimal policies. Before discussing these statistical properties, we introduce some notations and auxiliary quantities that facilitate our analysis.

Under Assumption 1, it is useful to consider the *span semi-norm* (Puterman, 1994). For vector $v \in V = \mathbb{R}^d$, let 1 be the vector with all entries equal to one and define

$$
\begin{aligned}
|v|_{\text{span}} &:= \inf_{c \in \mathbb{R}} \|v - c1\|_\infty \\
&= \max_{1 \le i \le d} v_i - \min_{1 \le i \le d} v_i.
\end{aligned}
\tag{A.1}
$$

Note that the span semi-norm satisfies the triangle inequality $|v + w|_{\text{span}} \le |v|_{\text{span}} + |w|_{\text{span}}$.

The analysis in this section will make use of the following standard deviation parameters: define

$$
\sigma(v)(s,a) := \sqrt{p_{s,a}[v^2] - p_{s,a}[v]^2}, \quad \text{and} \quad \sigma_\pi(v)(s) := \sigma(v)(s, \pi(s)).
\tag{A.2}
$$

Let $\pi_0$ be an optimal policy associated with MDPs $\mathcal{M}(\gamma, P, R)$. Recall that $\hat{\pi}_0$ is optimal for $\mathcal{M}(\gamma, \widehat{P}, R)$. Define for any $\pi \in \Pi$, $v_0^\pi := (I - \gamma P_\pi)^{-1} R_\pi$ and $\hat{v}_0^\pi := (I - \gamma \widehat{P}_\pi)^{-1} R_\pi$. Also, let $\hat{q}_0 = (I - \gamma \widehat{P}^{\hat{\pi}_0})^{-1} R_{\hat{\pi}_0}$.

Consider the following event

$$
\Omega_\eta := \left\{ \inf_{s \in S} \left( \hat{v}_0^{\hat{\pi}_0}(s) - \max_{b \ne \hat{\pi}_0(s)} \hat{q}_0(s,b) \right) \ge \eta \right\}.
\tag{A.3}
$$

This is a set of good events on which the optimality gap of MDP $\mathcal{M}(\gamma, \widehat{P}, R)$ is larger than $\eta$.

Recall the definition of the reward perturbation $Z$ in (3.1) and perturbed reward $R$. For any $\pi \in \Pi$, let $R_\pi(s) = R(s, \pi(s))$ and $Z_\pi(s) = Z(s, \pi(s))$ for all $s \in S$. To achieve the desired sample efficiency in terms of the minimum sample size, Li et al. (2020) recursively defines the auxiliary values:

$$
\begin{aligned}
h_0^\pi &= R_\pi; & v_0^\pi &= (I - \gamma P_\pi)^{-1} h_0^\pi; & \hat{v}_0^\pi &= (I - \gamma \widehat{P}_\pi)^{-1} h_0^\pi \\
h_l^\pi &= \sigma_\pi(v_{l-1}^\pi); & v_l^\pi &= (I - \gamma P_\pi)^{-1} h_l^\pi; & \hat{v}_l^\pi &= (I - \gamma \widehat{P}_\pi)^{-1} h_l^\pi
\end{aligned}
\tag{A.4}
$$

for all $l \ge 1$. Using these sequences and the techniques in Wang et al. (2023), we are able to show the following concentration bound:

**Proposition A.1.** *Assume Assumption 1 and for some $\eta \le 1$, $P(\Omega_\eta) \ge 1 - \delta/3$. For $n \ge 64\beta_\delta(\eta)(1-\gamma)^{-1}$, then w.p. at least $1 - \delta$ under $P$*

$$
\left\| \hat{v}_0^{\hat{\pi}_0} - v_0^{\hat{\pi}_0} \right\|_\infty \le 243 \sqrt{\frac{\beta_\delta(\eta) t_{\text{minorize}}}{(1-\gamma)^2 n}}, \quad \text{and} \quad v_0^{\pi_0} - v_0^{\hat{\pi}_0} \le 486 \sqrt{\frac{\beta_\delta(\eta) t_{\text{minorize}}}{(1-\gamma)^2 n}}
$$

*where*

$$
\beta_\delta(\eta) = 2 \log \left( \frac{24 |S||A| \log_2((1-\gamma)^{-1})}{(1-\gamma)^2 \eta \delta} \right).
$$

The proof of Proposition A.1 is provided in section C.2. We see that the conclusion of A.1 will resemble the statement of Theorem 1 if $P(\Omega_\eta) \ge 1 - \delta/3$ as well as $v_0^{\pi_0} \approx v^*$ and $v_0^{\hat{\pi}_0} \approx v^{\hat{\pi}_0}$ are sufficiently close. The following proposition in Li et al. (2020) proves the former requirement.

**Proposition A.2** (Lemma 6, Li et al. (2020)). *Let*

$$
\eta_\delta^* = \frac{\zeta \delta (1-\gamma)}{9|S||A|^2},
$$

*then $P(\Omega_{\eta_\delta^*}) \ge 1 - \delta/3$.*

## A.1 Proof of Theorem 1

Now, we are ready to present of proof of Theorem 1 given Propositions A.1 and A.2.

*Proof.* Observe that for any $\pi \in \Pi$, by the construction in (A.4),

$$v_0^\pi = (I - \gamma P_\pi)^{-1} R_\pi = v^\pi + (I - \gamma P_\pi)^{-1} Z_\pi.$$

Therefore,

$$\|v_0^\pi - v^\pi\|_\infty \le \zeta \left\|(I - \gamma P_\pi)^{-1}\right\|_{\infty,\infty} \le \frac{\zeta}{1 - \gamma}.$$

Consider

$$v^* - v^{\hat{\pi}_0} = (v^* - v_0^{\pi^*}) + (v_0^{\pi^*} - v_0^{\pi_0}) + (v_0^{\pi_0} - v_0^{\hat{\pi}_0}) + (v_0^{\hat{\pi}_0} - v^{\hat{\pi}_0})$$

$$\le \frac{2\zeta}{1 - \gamma} + v_0^{\pi_0} - v_0^{\hat{\pi}_0}$$

where the optimality of $\pi_0$ implies that $v_0^{\pi^*} - v_0^{\pi_0} \ge 0$. By Proposition A.1 and A.2, w.p. at least $1 - \delta$

$$v_0^{\pi_0} - v_0^{\hat{\pi}_0} \le 486 \sqrt{\frac{\beta_\delta(\eta_\delta^*) t_{\text{minorize}}}{(1 - \gamma)^2 n}}.$$

provided that $n \ge 64\beta_\delta(\eta_\delta^*)(1 - \gamma)^{-1}$.

To arrive at the sample complexity bound, we first note that $\beta_\delta(\eta_\delta^*)$ has log dependence on $\zeta$. Choose $\zeta = (1 - \gamma)\epsilon/4$ and for $c = 4 \cdot 486^2$,

$$n = \frac{c t_{\text{minorize}} \beta_\delta(\eta_\delta^*)}{(1 - \gamma)^2 \epsilon^2} = \widetilde{O}\left(\frac{t_{\text{minorize}}}{(1 - \gamma)^2 \epsilon^2}\right).$$

Then for $\epsilon \le \sqrt{t_{\text{minorize}}(1 - \gamma)^{-1}}$, $n \ge 64\beta_\delta(\eta_\delta^*)(1 - \gamma)^{-1}$ and w.p. at least $1 - \delta$

$$v^* - v^{\hat{\pi}_0} \le \frac{\epsilon}{2} + \frac{486\epsilon}{\sqrt{c}} = \epsilon.$$

$\square$

## B Reduction Bound and Optimal Sample Complexity for AMDP

This this section, we prove Theorem 2 given the DMDP optimal sample complexity result in Theorem 1. To achieve this, we need the following lemma that allows us to compare the long run average value and the discounted value of the MDP. From Lemma 3 Jin & Sidford (2021) and Theorem 1 of Wang et al. (2023) (c.f. equation (2.3)), we have that

**Lemma 1.** *Under Assumption 1, for all $\pi \in \Pi$,*

$$\|(1 - \gamma)v^\pi - \alpha^\pi\|_\infty \le 9(1 - \gamma)t_{\text{minorize}}.$$

### B.1 Proof of Theorem 2

Given Lemma 1 and Theorem 1, we present the proof of Theorem 2.

*Proof of Theorem 2.* First, note that $(1 - \gamma)^{-1} \ge t_{\text{minorize}} \ge 1$. By Theorem 1 and the choice of $\zeta$ and $n$, the policy $\hat{\pi}_0$ satisfies

$$v^* - v^{\hat{\pi}_0} \le t_{\text{minorize}}$$

w.p. at least $1 - \delta$.

Let $\bar{\pi}$ denote an optimal policy of the average reward problem. Then, w.p. at least $1 - \delta$,

$$\bar{\alpha} - \alpha^{\hat{\pi}_0} = [\bar{\alpha} - (1-\gamma)v^{\bar{\pi}}] + (1-\gamma)[v^{\bar{\pi}} - v^*] + (1-\gamma)[v^* - v^{\hat{\pi}_0}] + [(1-\gamma)v^{\hat{\pi}_0} - \alpha^{\hat{\pi}_0}]$$

$$\overset{(i)}{\leq} [\bar{\alpha} - (1-\gamma)v^{\bar{\pi}}] + (1-\gamma)[v^* - v^{\hat{\pi}_0}] + [(1-\gamma)v^{\hat{\pi}_0} - \alpha^{\hat{\pi}_0}]$$

$$\overset{(ii)}{\leq} 9(1-\gamma)t_{\mathrm{minorize}} + (1-\gamma)t_{\mathrm{minorize}} + 9(1-\gamma)t_{\mathrm{minorize}}$$

$$\overset{(iii)}{\leq} \epsilon$$

where $(i)$ uses the optimality of $\pi^*$, $(ii)$ follows from Lemma 1 and the $t_{\mathrm{minorize}}$-optimality of $\hat{\pi}_0$, $(iii)$ is due to the choice $1 - \gamma = \frac{\epsilon}{19t_{\mathrm{minorize}}}$. The total number of samples used is

$$|S||A|n = \tilde{O}\left(\frac{|S||A|}{(1-\gamma)^2 t_{\mathrm{minorize}}}\right) = \tilde{O}\left(\frac{|S||A|t_{\mathrm{minorize}}}{\epsilon^2}\right).$$

This implies the statement of Theorem 2. $\qquad\square$

## C  PROOFS OF KEY PROPOSITIONS

### C.1  DECOUPLING THE DEPENDENCE OF $\hat{P}$ AND $\hat{\pi}_0$

In this section, we introduce the method proposed by Agarwal et al. (2020) to decouple the statistical dependence of $\hat{p}_{s',a'}$ and $\hat{\pi}_0$ at a particular state action pair $(s', a') \in S \times A$. To simplify notation, we use $z = (s', a') \in S \times A$ to denote a pair of state and action. Define the kernel $\hat{K}^{(z)} := \left\{\hat{\kappa}_{s,a}^{(z)} \in \mathcal{P}(S) : (s,a) \in S \times A\right\}$ s.t. $\hat{\kappa}_{s',a'}^{(z)} = \delta_{s'}$ and $\hat{\kappa}_{s,a}^{(z)} = \hat{p}_{s,a}$ for all $(s,a) \neq z$. Therefore, under $\hat{K}^{(z)}$ and at state action pair $z = (s', a')$, the MDP will transition to $s'$ w.p.1, while other transitions are done according to $\hat{P}$.

Now, for fixed $\rho \in \mathbb{R}$, define a modified reward

$$h^{(z,\rho)}(s,a) = \rho \mathbb{1}\{z = (s,a)\} + R\mathbb{1}\{z \neq (s,a)\}, \quad (s,a) \in S \times A.$$

Let $\hat{g}^{(z,\rho)}$ be the unique solution of the Bellman equation

$$g(s) = \max_{a \in A}\left(h^{(z,\rho)}(s,a) + \gamma\hat{\kappa}_{s,a}^{(z)}[g]\right).$$

Let $\hat{\chi}^{(z,\rho)}$ be any optimal policy associated with $\hat{g}^{(z,\rho)}$. Now notice that since $\hat{\kappa}_{s',a'}^{(z)} = \delta_{s'}$ is non-random. Thus, $\hat{g}^{(z,\rho)}$ and $\hat{\chi}^{(z,\rho)}$ are independent of $\hat{p}_z$. By the definition of the auxiliary value functions in expression (A.4) above, $v_l^{(z,\rho)} := v_l^{\hat{\chi}^{(z,\rho)}}$ is also independent of $\hat{p}_z$. Therefore, if we define $\mathcal{G}_z := \sigma(\hat{p}_{s,a} : (s,a) \neq z, R)$, then $v_l^{(z,\rho)}$ is measureable w.r.t. $\mathcal{G}_z$.

Therefore, we can decouple the dependence of $\hat{P}_{s,a}$ and $\hat{\pi}_0$ by replacing $\hat{\pi}_0$ with $\hat{\chi}(z,\rho)$ for some $\rho$ so that $\hat{\pi}_0 = \hat{\chi}(z,\rho)$ with high probability. To achieve this, we first prescribe a finite set (a $\xi$-net of the interval $[-(1-\gamma)^{-1}, (1-\gamma)^{-1}]$ that will be denoted by $\mathcal{N}_\xi$) of possible values for $\rho$, and prove that for sufficiently small $\xi$, such a $\rho$ can be picked from this set. Note that the motivation for constructing $\mathcal{N}_\zeta$ and seeking a $\rho$ within it stems from the finite nature of this set, which enables us to apply the union bound technique.

Concretely, define an $\xi$-net on $[-(1-\gamma)^{-1}, (1-\gamma)^{-1}]$ by

$$\mathcal{N}_\xi := \{-k_\xi\xi, -(k_\xi - 1)\xi, \ldots, 0, \ldots, (k_\xi - 1)\xi, k_\xi\xi\}$$

where $k_\xi = \lfloor(1-\gamma)^{-1}\xi^{-1}\rfloor$. Note that, $|\mathcal{N}_\xi| \leq 2(1-\gamma)^{-1}\xi^{-1}$. Then the following lemma in Li et al. (2020) indicates that it is sufficient to set $\xi = (1-\gamma)\eta/4$ where we recall that $\eta$ is the optimality gap parameter of the event $\Omega_\eta$ in (A.3).

**Lemma 2** (Li et al. (2020), Lemma 4). *For each $z \in S \times A$, there exists $\rho_z \in \mathcal{N}_{(1-\gamma)\eta/4}$ s.t. $\hat{\chi}(z, \rho_z)(\omega) = \hat{\pi}_0(\omega)$ for all $\omega \in \Omega_\eta$.*

With this decoupling technique and policies $\hat{\chi}(z, \rho_z)$, we can approximate $v_l^{\hat{\pi}_0}$ by $v_l^{(z,\rho)}$ with $\rho_z \in \mathcal{N}_{(1-\gamma)\eta/4}$. In particular, the following concentration inequality for $v_l^{(z,\rho)}$ can be translated to that of $v_l^{\hat{\pi}_0}$, leading to our proof of Proposition A.1.

**Lemma 3** (Bernstein's Inequality). *For each $z \in S \times A$, consider any finite set $U^{(z)} \subset \mathbb{R}$, then w.p. at least $1 - \delta$ under $P$, we have that for all $0 \le l \le l^*$, $z \in S \times A$, $\rho \in U^{(z)}$*

$$\left| (\hat{p}_z - p_z) \left[ v_l^{(z,\rho)} \right] \right| \le \sqrt{\frac{\beta}{n}} \sigma(v_l^{(z,\rho)})(z) + \frac{\beta}{n} \left| v_l^{(z,\rho)} \right|_{\text{span}}$$

*where*

$$\beta := 2 \log \left( \frac{2|S||A|l^*|U|}{\delta} \right)$$

*and $|U| := \sup_{z \in S \times A} |U^{(z)}|$.*

This lemma is proved in Appendix D.1. We note that, of course, $\mathcal{N}_{(1-\gamma)\eta/4}$ will be used in place of $U^{(z)}$.

## C.2 PROOF OF PROPOSITION A.1

Before we prove Proposition A.1, we introduce the following lemma that converts Bernstein-type inequalities for $\{v_l\}$ as in Lemma 3 to concentration bound of $\hat{v}_0$.

**Lemma 4.** *Fix any $b > 0$ and possibly data dependent policy $\tilde{\pi}(\omega) \in \Pi$. Let $B \subset \Omega$ be s.t. $\forall \omega \in B$, the sequence $\{v_l^{\tilde{\pi}}\}$ defined in (A.4) satisfies*

$$\left| (\hat{P}_{\tilde{\pi}} - P_{\tilde{\pi}}) v_l^{\tilde{\pi}} \right| \le \sqrt{\frac{b}{n}} \sigma_{\tilde{\pi}}(v_l^{\tilde{\pi}}) + \frac{b}{n} \left| v_l^{\tilde{\pi}} \right|_{\text{span}}$$

*for all $l \le l^* = \left\lfloor \frac{1}{2} \log_2((1-\gamma)^{-1}) \right\rfloor$, where the absolute value is taken entry-wise. Then*

$$\left\| \hat{v}_0^{\tilde{\pi}} - v_0^{\tilde{\pi}} \right\|_\infty \le 243 \sqrt{\frac{b t_{\text{minorize}}}{(1-\gamma)^2 n}}$$

*on $B$, provided that $n \ge 64b(1-\gamma)^{-1}$.*

The proof of this lemma is provided in Appendix D.2. Note the appearance of $t_{\text{minorize}}$ in the bound. This is a consequence of the analysis technique in Wang et al. (2023). With the above auxiliary definitions and results, we prove Proposition A.1

*Proof of Proposition A.1.* We proceed with proving the first bound. By Lemma 2 and 3 with $\delta$ replaced by $\delta/3$ and $U^{(z)}$ by $\mathcal{N}_{(1-\gamma)\eta/4}$, there exists $B \subset \Omega$ s.t. $P(B) \ge 1 - \delta/3$ and on $B \cap \Omega_\eta$

$$\left| (\hat{p}_z - p_z) \left[ v_l^{\hat{\pi}_0} \right] \right| = \left| (\hat{p}_z - p_z) \left[ v_l^{(z,\rho_z)} \right] \right|$$

$$\le \sqrt{\frac{\beta_\delta(\eta)}{n}} \sigma(v_l^{(z,\rho_z)})(z) + \frac{\beta_\delta(\eta)}{n} \left| v_l^{(z,\rho_z)} \right|_{\text{span}}$$

$$\le \sqrt{\frac{\beta_\delta(\eta)}{n}} \sigma(v_l^{\hat{\pi}_0})(z) + \frac{\beta_\delta(\eta)}{n} \left| v_l^{\hat{\pi}_0} \right|_{\text{span}}.$$

for all $0 \le l \le l^* = \left\lfloor \frac{1}{2} \log_2((1-\gamma)^{-1}) \right\rfloor$, $z \in S \times A$, $\rho \in U^{(z)}$. In particular, on $B \cap \Omega_\eta$

$$\left| (\hat{P}_{\hat{\pi}_0} - P_{\hat{\pi}_0}) v_l^{\hat{\pi}_0} \right| \le \sqrt{\frac{\beta_\delta(\eta)}{n}} \sigma_{\hat{\pi}_0}(v_l^{\hat{\pi}_0}) + \frac{\beta_\delta(\eta)}{n} \left| v_l^{\hat{\pi}_0} \right|_{\text{span}}$$

all $0 \le l \le l^*$. Therefore, we conclude that by Lemma 4

$$\left\| \hat{v}_0^{\hat{\pi}_0} - v_0^{\hat{\pi}_0} \right\|_\infty \le 243 \sqrt{\frac{\beta_\delta(\eta) t_{\text{minorize}}}{(1-\gamma)^2 n}}$$

on $B \cap \Omega_\eta$. Sinece $P(\Omega_\eta) \geq 1 - \delta/3$, apply union bound, one obtains the first claimed.

To prove the second claim, we note that

$$0 \leq v_0^{\pi_0} - v_0^{\hat{\pi}_0}$$
$$= (v_0^{\pi_0} - \hat{v}_0^{\pi_0}) + (\hat{v}_0^{\pi_0} - \hat{v}_0^{\hat{\pi}_0}) + (\hat{v}_0^{\hat{\pi}_0} - v_0^{\hat{\pi}_0})$$
$$\leq \|v_0^{\pi_0} - \hat{v}_0^{\pi_0}\|_\infty + \left\|\hat{v}_0^{\hat{\pi}_0} - v_0^{\hat{\pi}_0}\right\|_\infty$$

where the last inequality follows from $\hat{v}_0^{\pi_0} - \hat{v}_0^{\hat{\pi}_0} \leq 0$. Hence, it remains to bound $\left\|v^* - \hat{v}^{\pi^*}\right\|_\infty$.

It is easy to see that the same proof of Lemma 3 implies that

$$\left|(\widehat{P}_{\pi_0} - P_{\pi_0})v_l^{\pi_0}\right| \leq \sqrt{\frac{\beta_\delta(\eta)}{n}}\sigma_{\pi_0}(v_l^{\pi_0}) + \frac{\beta_\delta(\eta)}{n}\left|v_l^{\pi_0}\right|_{\text{span}}$$

for all $0 \leq l \leq l^*$ w.p. at least $1 - \delta/3$. Therefore, by Lemma 4,

$$\|v_0^{\pi_0} - \hat{v}_0^{\pi_0}\|_\infty \leq 243\sqrt{\frac{\beta_\delta(\eta)t_{\text{minorize}}}{(1-\gamma)^2 n}}.$$

w.p. at least $1 - \delta/3$. Again, an application of the union bound completes the proof. □

## D  Proofs of Auxiliary Lemmas

### D.1  Proof of Lemma 3

*Proof.* Recall that $\mathcal{G}_z := \sigma(\hat{p}_{s,a} : (s,a) \neq z, R)$. For each $z \in S \times A$, define the probability measure $P_z(\cdot) := P(\cdot|\mathcal{G}_z)$ and expectation $E_z[\cdot] := E[\cdot|\mathcal{G}_z]$. Fix any $0 \leq l \leq l^*$ and $\rho \in U^{(z)}$. Since $(\hat{p}_z - p_z)[1] = 0$,

$$\left|(\hat{p}_z - p_z)\left[v_l^{(z,\rho)}\right]\right| \leq 2\left|v_l^{(z,\rho)}\right|_{\text{span}}.$$

As $\hat{p}_z$ is independent of $\mathcal{G}_z$, and $v_l^{(z,\rho)}$ is measureable w.r.t. $\mathcal{G}_z$,

$$E_z\left[(\hat{p}_z - p_z)\left[v_l^{(z,\rho)}\right]\right] = 0.$$

Therefore, by Bernstein's inequality, for any $t \geq 0$

$$P_z\left(\left|(\hat{p}_z - p_z)\left[v_l^{(z,\rho)}\right]\right| > t\right) \leq 2\exp\left(-\frac{n^2 t^2}{2n\sigma^2(v_l^{(z,p)})(z) + \frac{4}{3}\left|v_l^{(z,\rho)}\right|_{\text{span}} nt}\right).$$

Choose $t$ s.t. the r.h.s. is less than $\delta/(|S||A|l^*|U|)$, we find that it is sufficient for

$$t \geq \sqrt{\frac{2\log\left(\frac{2|S||A|l^*|U|}{\delta}\right)}{n}}\sigma(v_l^{(z,p)})(z) + \frac{4}{3}\log\left(\frac{2|S||A|l^*|U|}{\delta}\right)\frac{\left|v_l^{(z,\rho)}\right|_{\text{span}}}{n}.$$

Clearly

$$t_0 := \sqrt{\frac{\beta}{n}}\sigma(v_l^{(z,\rho)}) + \frac{\beta}{n}\left|v_l^{(z,\rho)}\right|_{\text{span}}$$

satisfies the above inequality. Therefore,

$$P\left(\max_{l \leq l^*, z \in S \times A}\max_{\rho \in U^{(z)}}\left|(\hat{p}_z - p_z)\left[v_l^{(z,\rho)}\right]\right| > t_0\right)$$
$$\leq \sum_{l \leq l^*, z \in S \times A}\sum_{\rho \in U^{(z)}} EP_z\left(\left|(\hat{p}_z - p_z)\left[v_l^{(z,\rho)}\right]\right| > t_0\right)$$
$$\leq \sum_{l \leq l^*, z \in S \times A}\sum_{\rho \in U^{(z)}}\frac{\delta}{|S||A|l^*|U|}$$
$$\leq \delta$$

This directly implies the statement of the lemma. □

## D.2 PROOF OF LEMMA 4

*Proof.* First, observe that by the definition in (A.4),

$$
\begin{aligned}
\hat{v}_l^{\tilde{\pi}} - v_l^{\tilde{\pi}} &= (I - \gamma\widehat{P}_{\tilde{\pi}})^{-1}h_l^{\tilde{\pi}} - (I - \gamma\widehat{P}_{\tilde{\pi}})^{-1}(I - \gamma\widehat{P}_{\tilde{\pi}})v_l^{\tilde{\pi}} \\
&= (I - \gamma\widehat{P}_{\tilde{\pi}})^{-1}(I - \gamma P_{\tilde{\pi}})v_l^{\tilde{\pi}} - (I - \gamma\widehat{P}_{\tilde{\pi}})^{-1}(I - \gamma\widehat{P}_{\tilde{\pi}})v_l^{\tilde{\pi}} \\
&= \gamma(I - \gamma\widehat{P}_{\tilde{\pi}})^{-1}(\widehat{P}_{\tilde{\pi}} - P_{\tilde{\pi}})v_l^{\tilde{\pi}}
\end{aligned}
$$

Let us define $\Delta_l := \left\|\hat{v}_l^{\tilde{\pi}} - v_l^{\tilde{\pi}}\right\|_\infty$. By the assumption of Lemma 4, for all $0 \le l \le l^*$, on the event $B$ in the lemma,

$$
\begin{aligned}
\Delta_l &\overset{(i)}{\le} \left\|\gamma(I - \gamma\widehat{P}_{\tilde{\pi}})^{-1}\left|(\widehat{P}_{\tilde{\pi}} - P_{\tilde{\pi}})v_l^{\tilde{\pi}}\right|\right\|_\infty \\
&\overset{(ii)}{\le} \gamma\sqrt{\frac{b}{n}}\left\|(I - \gamma\widehat{P}_{\tilde{\pi}})^{-1}\sigma_{\tilde{\pi}}(v_l^{\tilde{\pi}})\right\|_\infty + \frac{\gamma b}{(1-\gamma)n}\left|v_l^{\tilde{\pi}}\right|_{\mathrm{span}} \\
&= \gamma\sqrt{\frac{b}{n}}\left\|\hat{v}_{l+1}^{\tilde{\pi}}\right\|_\infty + \frac{\gamma b}{(1-\gamma)n}\left|v_l^{\tilde{\pi}}\right|_{\mathrm{span}} \\
&\le \gamma\sqrt{\frac{b}{n}}\Delta_{l+1} + \gamma\sqrt{\frac{b}{n}}\left\|v_{l+1}^{\tilde{\pi}}\right\|_\infty + \frac{\gamma b}{(1-\gamma)n}\left|v_l^{\tilde{\pi}}\right|_{\mathrm{span}}
\end{aligned}
$$

(D.1)

where $(i)$ and $(ii)$ follow from $(I - \gamma\widehat{P}_{\tilde{\pi}})^{-1}$ being non-negative so that $(I - \gamma\widehat{P}_{\pi})^{-1}h \le (I - \gamma\widehat{P}_{\pi})^{-1}|h| \le (I - \gamma\widehat{P}_{\pi})^{-1}g$ for all function $h : S \to \mathbb{R}$ and $g \ge |h|$.

We can think of (D.1) as a recursive bound for $\Delta_0$. To analyze this recursive bound, we first consider the following. By Lemma 11 of Li et al. (2020), we have that for $l \ge 0$

$$
\begin{aligned}
\left\|v_{l+1}^{\tilde{\pi}}\right\|_\infty &= \left\|(I - \gamma P_{\tilde{\pi}})^{-1}\sigma_{\tilde{\pi}}(v_l^{\tilde{\pi}})\right\|_\infty \\
&\le \frac{4}{\gamma\sqrt{1-\gamma}}\left\|v_l^{\tilde{\pi}}\right\|_\infty \\
&\le \cdots \\
&\le \left(\frac{4}{\gamma\sqrt{1-\gamma}}\right)^l \left\|v_1^{\tilde{\pi}}\right\|_\infty
\end{aligned}
$$

(D.2)

Therefore, expanding the recursion (D.1)

$$
\begin{aligned}
\Delta_0 &\le \gamma\sqrt{\frac{b}{n}}\Delta_1 + \gamma\sqrt{\frac{b}{n}}\left\|v_1^{\tilde{\pi}}\right\|_\infty + \frac{\gamma b}{(1-\gamma)n}\left|v_0^{\tilde{\pi}}\right|_{\mathrm{span}} \\
&\le \cdots \\
&\le \left(\gamma\sqrt{\frac{b}{n}}\right)^{l^*}\Delta_{l^*} + \sum_{k=1}^{l^*}\left(\gamma\sqrt{\frac{b}{n}}\right)^k\left\|v_k^{\tilde{\pi}}\right\|_\infty + \frac{\gamma b}{(1-\gamma)n}\sum_{k=0}^{l^*-1}\left(\gamma\sqrt{\frac{b}{n}}\right)^k\left|v_k^{\tilde{\pi}}\right|_{\mathrm{span}}
\end{aligned}
$$

Since $v_k^{\tilde{\pi}} \ge 0$ for all $k \ge 0$, $\left|v_k^{\tilde{\pi}}\right|_{\mathrm{span}} \le \left\|v_k^{\tilde{\pi}}\right\|_\infty$. We have that

$$
\begin{aligned}
\Delta_0 &\le \left(\gamma\sqrt{\frac{b}{n}}\right)^{l^*}\Delta_{l^*} + \left(1 + \frac{\gamma b}{(1-\gamma)n}\right)\sum_{k=1}^{l^*}\left(\gamma\sqrt{\frac{b}{n}}\right)^k\left\|v_k^{\tilde{\pi}}\right\|_\infty + \frac{\gamma b}{(1-\gamma)n}\left|v_0^{\tilde{\pi}}\right|_{\mathrm{span}} \\
&=: E_1 + E_2 + E_3
\end{aligned}
$$

(D.3)

We first consider $E_1$. By the identities (D.1) and (D.2)

$$
\begin{aligned}
\Delta_{l^*} &\leq \gamma\sqrt{\frac{b}{n}}\left\|(I-\gamma\widehat{P}_{\tilde{\pi}})^{-1}\sigma_{\tilde{\pi}}(v_l^{\tilde{\pi}})\right\|_{\infty} + \frac{\gamma b}{(1-\gamma)n}\left|v_{l^*}^{\tilde{\pi}}\right|_{\text{span}} \\
&\leq \frac{\gamma}{1-\gamma}\sqrt{\frac{b}{n}}\left\|\sigma_{\tilde{\pi}}(v_l^{\tilde{\pi}})\right\|_{\infty} + \frac{\gamma b}{(1-\gamma)n}\left|v_{l^*}^{\tilde{\pi}}\right|_{\text{span}} \\
&\leq \left(\frac{\gamma}{(1-\gamma)}\sqrt{\frac{b}{n}} + \frac{\gamma b}{(1-\gamma)n}\right)\left\|v_{l^*}^{\tilde{\pi}}\right\|_{\infty} \\
&\leq \left(\frac{\gamma}{(1-\gamma)}\sqrt{\frac{b}{n}} + \frac{\gamma b}{(1-\gamma)n}\right)\left(\frac{4}{\gamma\sqrt{1-\gamma}}\right)^{l^*-1}\left\|v_1^{\tilde{\pi}}\right\|_{\infty} \\
&\leq \frac{\gamma^2}{4}\left(\sqrt{\frac{b}{(1-\gamma)n}} + \frac{b}{(1-\gamma)n}\right)\left(\frac{4}{\gamma\sqrt{1-\gamma}}\right)^{l^*}\left\|v_1^{\tilde{\pi}}\right\|_{\infty} \\
&\overset{(i)}{\leq} \frac{\gamma^2}{2}\sqrt{\frac{b}{(1-\gamma)n}}\left(\frac{4}{\gamma\sqrt{1-\gamma}}\right)^{l^*}\left\|v_1^{\tilde{\pi}}\right\|_{\infty}
\end{aligned}
$$

where $(i)$ follows from $n \geq b/(1-\gamma)$. Therefore,

$$
\begin{aligned}
E_1 &\leq \frac{\gamma^2}{2\sqrt{1-\gamma}}\left(\frac{16b}{(1-\gamma)n}\right)^{(l^*+1)/2}\sqrt{\frac{b}{n}}\left\|v_1^{\tilde{\pi}}\right\|_{\infty} \\
&\overset{(i)}{\leq} \frac{1}{\sqrt{1-\gamma}}2^{-(l^*+2)}\sqrt{\frac{b}{n}}\left\|v_1^{\tilde{\pi}}\right\|_{\infty} \\
&\overset{(ii)}{\leq} \frac{1}{\sqrt{1-\gamma}}2^{\log_2(1-\gamma)/2}\sqrt{\frac{b}{n}}\left\|v_1^{\tilde{\pi}}\right\|_{\infty} \\
&\leq \sqrt{\frac{b}{n}}\left\|v_1^{\tilde{\pi}}\right\|_{\infty}
\end{aligned}
$$

where $(i)$ follows from the assumption that $n \geq 64b(1-\gamma)^{-1}$ and $(ii)$ is due to $l^*+2 \geq \frac{1}{2}\log_2((1-\gamma)^{-1})$.

Next, we bound $E_2$. By (D.2)

$$
\begin{aligned}
E_2 &\leq \frac{\gamma\sqrt{1-\gamma}}{2}\sum_{k=1}^{l^*+1}\left(\gamma\sqrt{\frac{b}{n}}\right)^k\left(\frac{4}{\gamma\sqrt{1-\gamma}}\right)^k\left\|v_1^{\tilde{\pi}}\right\|_{\infty} \\
&\leq 2\gamma\sqrt{\frac{b}{n}}\left\|v_1^{\tilde{\pi}}\right\|_{\infty}\sum_{k=0}^{\infty}\left(\sqrt{\frac{16b}{(1-\gamma)n}}\right)^k \\
&\leq 2\sqrt{\frac{b}{n}}\left\|v_1^{\tilde{\pi}}\right\|_{\infty}.
\end{aligned}
$$

Also, note that $v_0^{\tilde{\pi}} = (I-\gamma P_{\tilde{\pi}})^{-1}r_{\tilde{\pi}} = v^{\tilde{\pi}}$ and $v_1^{\tilde{\pi}} = (I-\gamma P_{\tilde{\pi}})^{-1}\sigma_{\tilde{\pi}}(v^{\tilde{\pi}})$. By Proposition 6.1 and Corollary 6.2.1 of Wang et al. (2023)

$$
\left|v_0^{\tilde{\pi}}\right|_{\text{span}} \leq 3t_{\text{minorize}} \quad \text{and} \quad \left\|v_1^{\tilde{\pi}}\right\|_{\infty} \leq 80\frac{\sqrt{t_{\text{minorize}}}}{1-\gamma}.
$$

Thus we conclude that

$$
\begin{aligned}
\Delta_0 &\leq 3\sqrt{\frac{b}{n}}\left\|v_1^{\tilde{\pi}}\right\|_{\infty} + \frac{\gamma b}{(1-\gamma)n}\left|v_0^{\tilde{\pi}}\right|_{\text{span}} \\
&\leq \frac{1}{1-\gamma}\left(240\sqrt{\frac{bt_{\text{minorize}}}{n}} + \frac{3bt_{\text{minorize}}}{n}\right) \\
&\leq 243\sqrt{\frac{bt_{\text{minorize}}}{(1-\gamma)^2 n}}
\end{aligned}
$$

where the last inequality follows from $t_{\mathrm{minorize}} \leq (1-\gamma)^{-1}$ and so $bt_{\mathrm{minorize}}/n \leq 1$ by assumption on $n$. $\qquad\square$

