# OpenReview forum: "Optimal Sample Complexity for Average Reward Markov Decision Processes"
_ICLR.cc/2024/Conference — ICLR 2024 poster_

### Official Review · Reviewer_iFE8 · 2023-10-22

**Soundness:** 3 good
**Presentation:** 3 good
**Contribution:** 3 good
**Rating:** 6
**Confidence:** 3

**Summary:**

This paper studies the sample complexity of learning the optimal policy in average-reward Markov decision processes, under the assumption of a uniformly ergodic MDP and a generative model. The proposed algorithm improves the best sample complexity upper bound in existing works and matches the lower bound of the problem. This is achieved by combining the algorithmic ideas of two lines of existing research, by first reducing the AMDP to a discounted-reward MDP and then establishing an optimal sample complexity upper bound in the setting of uniformly ergodic discounted MDPs.

**Strengths:**

This paper studies an important problem in reinforcement learning theory. It closes the gap between the upper and lower bounds of the sample complexity for learning an optimal policy. The technical proofs look solid as far as I can tell. The presentation of the algorithm and analysis is also clear.

**Weaknesses:**

As the authors have mentioned in the paper, the algorithm is developed by combining the algorithmic ideas from two lines of existing research (Jin & Sidford, 2021) and (Li et al., 2020). While in general we would hope to see technical novelty in terms of algorithm design, it is probably okay with this work because reducing to a discounted MDP (Jin & Sidford, 2021) seems to be a standard approach, and the authors do make improvements over the analysis of (Li et al., 2020) to establish a sharper sample complexity upper bound.

Even though I understand that the main contributions of this work are theoretical, I would still hope to see some numerical results to demonstrate some of the ideas in the paper.

**Questions:**

Does your work imply any new results for the case where the uniform ergodicity assumption does not hold, such as the weakly communicating AMDP setting?

---

> ### Author Response · Authors · 2023-11-17
>
> Thanks for your comments and the question! We hope that the following will resolve some of your concerns:
> 1. Reduction to AMDP approach: The idea of solving an AMDP by reducing it to a DMDP and applying dynamic programming has been considered and implemented since the 1970s, see Hordijk and Tijms (1975). In fact, the optimality of our DMDP algorithm implies that one can choose $\gamma$ of the approximating DMDP to be larger than that in Jin and Sidford (2021), where $(1-\gamma) = \Theta(\epsilon/t_{mix})$, and still get the optimal sample complexity. This is not the case in Jin and Sidford (2021), where enlarging $\gamma$ will worsen the sample complexity. However, note that choosing large $\gamma$ is inefficient from a computational standpoint, because solving the empirical MDP with $\gamma$ close to 1 is typically hard (consider using value iteration). Thus, we choose $\gamma$ to be as small as possible, yielding the same $(1-\gamma) = \Theta(\epsilon/t_{mix})$ as in Jin and Sidford (2021).
> 2. Numerical experiments: Thanks for the suggestion. We added two experiments that separately verify our algorithm's complexity dependence on $\epsilon$ and $t_{mix}$, yielding results that are consistent with our theory's prediction.
> 3. The weakly communicating AMDP setting: We are not aware of a way to generalize our method to the weak communication setting that can improve the existing sample complexity dependence of $\widetilde O (H^2\epsilon^{-2})$ in Zhang and Xie (2023). The weak communication setting is connected to a simpler DMDP setting where only one optimal policy induces mixing. In this simpler setting, there is no known matching sample complexity upper and lower bound in the literature, while an upper bounds with $\widetilde O(t_{mix}^2\epsilon^{-2})$ dependence is achieved in Wang et al. (2023). We believe that a variant of our method can yield a $\widetilde O (H^2\epsilon^{-2})$ dependence, matching that in Zhang and Xie (2023). But we are not sure how to improve upon this.

---

### Official Review · Reviewer_tPAf · 2023-10-29

**Soundness:** 3 good
**Presentation:** 3 good
**Contribution:** 3 good
**Rating:** 6
**Confidence:** 3

**Summary:**

The authors considered the sample complexity of average reward MDPs under the uniformly ergodic condition, and provided a novel analysis for the algorithm given in Li et al. 2020, which results in an upper bound that matches the known lower bound.

**Strengths:**

1. Theoretical paper that gives a matching bound, therefore fully establishing the optimal sample complexity for AMDP under the uniform ergodicity condition.
2. The background was explained clearly, and the context of the result to other related settings is also well explained.
3. In most places, the notations and proofs are done rigorously, more so than the average papers.

**Weaknesses:**

1. The result is somewhat thin in the sense that it feels like filling a small gap that was somehow overlooked by several previous groups of researchers, though I personally like the cleanness of the result.
2. The main contribution is technical, yet the main paper does not really spend the effort to clearly explain the technical critical point that enables the authors to establish the bound. Particularly, it appears Proposition A.1 is the critical step to establish the concentration inequality. More discussion on the technical level on the difference with previous bounds should be given.

**Questions:**

1. I'm a little confused about some notation. Is alpha in (2.6) in R^|S|? It should be, but later in the definition of \bar{alpha}, we need to choose the maximum alpha, which seems incorrect since it is a vector.
2. Equation (2.4), where is \eta defined? It seems to appear without any context.
3. Can you comment on the choice of distribution of Z(s,a)? Does it make a difference if another distribution (non-uniform) is used?
4. Though I can understand the result is theoretical, have the authors used any numerical results to verify the optimal algorithm behavior that suggests the sample complexity scaling?

---

> ### Author Response · Authors · 2023-11-17
>
> Thanks for the careful reading and suggestions! We hope that the following can resolve your concerns and questions:
> 1. Contribution: The optimal sample complexity for AMDP has been recognized as a significant and challenging open problem in RL theory, motivating many attempts and improvements over the last 6 years (see Table 1). Our algorithm is the first one in the literature that achieves minmax optimal sample complexity for uniformly ergodic MDP. This is achieved by bringing in multiple algorithmic ideas and analysis techniques. We highlight this contribution in our revised abstract and in the third paragraph of the introduction.
> 2. Technical challenge: The main challenge to our algorithm design and analysis is to achieve a $\widetilde O(t_{mix}(1-\gamma)^{-2}\epsilon^{-2})$ sample complexity with small minimum sample size. This is explained in the introduction, Sections 3, and 3.1. These are resolved in Theorem 1 by the error upper bound (3.1) where we obtain a term of the form $\sqrt{t_{mix}(1-\gamma)^{-2} n^{-1}}$ and the minimum sample size $n = 64\beta (1-\gamma)^{-1}$.  We added a remark to highlight this. The error bound of $\sqrt{t_{mix}(1-\gamma)^{-2} n^{-1}}$ from Proposition A.1 requires a key step that decouples the dependence of $\widehat P$ and $\hat\pi_0$, an idea that is due to Agarwal et al. (2020). We include a detailed presentation of the procedure in Appendix C.
> 3. Definition of $\alpha^{\pi}$: The $\alpha^{\pi}$ is the long run average reward. In general, it is indeed a function $S\rightarrow R$. However, because we assume that all policies induce mixing (Assumption 1), the long-run average reward induced by any policy is independent of the initial state. Therefore, as explained between equations (2.3) and (2.4), we define it as a limit $\alpha^\pi\in R$.
> 4. $\eta(\cdot)$ definition: Thanks for pointing this out! We made changes to the manuscript, and now $\eta_\pi$ is the stationary distribution of $P_\pi$.
> 5. Uniform distribution of $Z$: $Z$ need not be uniformly distributed. It only serves as a perturbation so that the optimal value is unique with a positive optimality gap. This technique is introduced by Li et al. (2022). We added a short discussion about this after equation (3.1).
> 6. Numerical experiments: Thanks for the suggestion. We added two experiments that separately verify our algorithm's complexity dependence on $\epsilon$ and $t_{mix}$, yielding results that are consistent with our theory's prediction.

---

> > ### Comment · Reviewer_tPAf · 2023-11-23
> >
> > Thanks for the clarifications and the addition of the numerical results. I'm still not completely convinced of the significance of the result under this specific technical condition of uniform ergodicity. I'm keeping the score.

---

### Official Review · Reviewer_qCaw · 2023-10-30

**Soundness:** 4 excellent
**Presentation:** 3 good
**Contribution:** 3 good
**Rating:** 8
**Confidence:** 4

**Summary:**

The authors presents the sample complexity result for the average-reward Markov Decision Processes (AMDPs) under the assumption of ergodicity and with an access to a simulator. This sample complexity is nearly minimax optimal in the class of ergodic MDP, thus closing the gap in mixing time or desired accuracy that appears in previous works. The presented algorithm is basically applies the reduction technique from AMDP to DMDP and efficiently exploits ergodicity assumption in the DMDP setup.

**Strengths:**

- First minimax optimal guarantees for AMDPs under a generative model assumptions;
- As a byproduct, authors provide minimax optimal  for ergodic DMDPs
- Computationally feasible algorithm;
- Simplicity of the presented approach.

**Weaknesses:**

- All the main instruments has already introduced in other papers, and thus this paper may lack of novelty.
    - Reduction of AMDP to DMDP is presented in (Jin & Sidford, 2021)
    - Optimal rates with optimal warm-up are presented in (Li et al. 2020);
    - Rates for mixing DMDP are already presented in (Wang et al. 2023) (specifically, Proposition 6.1 and Corollary 6.2.1);

Yujia Jin and Aaron Sidford. Towards tight bounds on the sample complexity of average-reward
MDPs, 2021.

Gen Li, Yuting Wei, Yuejie Chi, Yuantao Gu, and Yuxin Chen. Breaking the sample size barrier
in model-based reinforcement learning with a generative model. In H. Larochelle, M. Ranzato,
R. Hadsell, M.F. Balcan, and H. Lin (eds.), Advances in Neural Information Processing Systems,
volume 33, pp. 12861–12872. Curran Associates, Inc., 2020.

Shengbo Wang, Jose Blanchet, and Peter Glynn. Optimal sample complexity of reinforcement
learning for uniformly ergodic discounted Markov decision processes, 2023.

**Questions:**

- What are main barriers to provide an algorithm with dependence not on a mixing time-type quantity but on span of optimal value? This questions has its importance because it is know that this guarantee will be strictly tighter than mixing dependent.
- Is it possible to provide an algorithm without a reduction to discounted setting?
- Is it possible to extend this approach to exploration setup and provide a feasible algorithm with guarantees like (Orther, 2020)?

Ortner, Ronald. "Regret bounds for reinforcement learning via markov chain concentration." *Journal of Artificial Intelligence Research* 67 (2020): 115-128.

---

> ### Author Response · Authors · 2023-11-17
>
> Thanks for your insightful questions. We hope that the following answers will clarify them, and we are happy to discuss them further.
> 1. Span semi norm $H$: As you suggest, the parameter $H$ has advantages over $t_{mix}$ as the basis for an upper bound. We included a short discussion of this in Section 5. Moreover, we have evidence suggesting that the natural analog of our algorithm should have a sample complexity upper bound of $\widetilde O (|S||A|H\epsilon^{-2})$, if we replace the uniform ergodicity assumption by one that assumes the span semi-norm of the transient value function induced by any policy is bounded by $H$. However, we didn't adopt this setup for the following reasons. First, $t_{mix}$ places a direct assumption on the primitives---namely the transition kernel---of the AMDP, yet the assumption on $H$ involves the solution to Poisson equations. Second, we are aiming at a "worst case" complexity result for the AMDP. The $H$ metric, considering the interplay between the reward and the transition kernel, has the flavor of instance-dependent complexity theory (see Khamaru et al. (2021), cited in the paper).
> 2. Reduction procedure: Thanks for asking about this. Before giving a direct response, we would like to point out that the idea of solving an AMDP by reducing it to a DMDP and applying dynamic programming has been considered and implemented since the 1970s, see Hordijk and Tijms (1975). In fact, the optimality of our DMDP algorithm implies that one can choose $\gamma$ of the approximating DMDP to be much smaller than that in Jin and Sidford (2021), where $(1-\gamma) = \Theta(\epsilon/t_{mix})$, and still get the optimal sample complexity. This is not the case in Jin and Sidford (2021), where enlarging $\gamma$ will worsen the sample complexity. However, note that choosing large $\gamma$ is inefficient from a computational standpoint, because solving the empirical MDP with large $\gamma$ is typically hard (consider using value iteration), and hence is not implemented in the paper. Therefore, the reduction method is mainly a computational tool. So, statistically speaking, we think it is possible to achieve the same sample complexity using a model-based approach without the reduction. However, this might lead to computational complications, e.g. slow convergence rate for non-mixing empirical AMDPs.
> 3. Extension to the online setting: The key ideas and bounds extend to the online setting without any issue. As our answer to your question (2) suggests, our bound is not reliant on a particular choice of the discount $\gamma$. Therefore, we do believe that our theory can be used to extend classical model-based online discounted RL algorithms to the average reward setting.

---

### Official Review · Reviewer_fwUc · 2023-10-31

**Soundness:** 3 good
**Presentation:** 3 good
**Contribution:** 2 fair
**Rating:** 6
**Confidence:** 2

**Summary:**

This paper resolves the issue of sample complexity associated with maximizing the long-term average reward dictated by a uniformly ergodic Markov Decision Process (MDP), predicated on the assumption of a generative model. The findings of this study enhance the pre-existing results by a factor of $t_{mix} and align with the established lower bound. The algorithm introduced herein is a synthesis of the methodologies proposed by Jin & Sidford (2021) and Li et al. (2020).

**Strengths:**

1. The paper addresses a significant issue in the domain of Markov Decision Processes, providing a solution to the sample complexity associated with maximizing the long-term average reward. This is a valuable contribution that could potentially advance understanding and application in this area.
2. By enhancing pre-existing results by a factor of the mixing time and aligning with the established lower bound, the paper provides a comparative analysis that underscores the improvements made and the relevance of the studies.
3. This paper is well-written and easy to understand.

**Weaknesses:**

1. The algorithm is primarily a synthesis of methodologies from Jin & Sidford (2021) and Li et al. (2020). While this approach has its merits, the novelty of this paper is somewhat limited given its dependence on previous works.
2. The paper could be enhanced by placing greater emphasis on the challenges addressed by the study and the innovative aspects of the proposed algorithm. Highlighting these elements would help to showcase the unique contributions of the paper and further establish its significance in the field.

**Questions:**

1. The paper could benefit from a greater emphasis on the challenges addressed by the study. Could you provide more information on the specific challenges inherent to the problem you are solving and how your approach effectively addresses these issues?
2. Could you please elaborate on the unique aspects of your algorithm and how it distinctly contributes to the field beyond the synthesis of methodologies from Jin & Sidford (2021) and Li et al. (2020)? Highlighting the novel components of your approach could significantly strengthen the impact of your paper.

---

> ### Author Response · Authors · 2023-11-17
>
> Thanks for your comments and suggestions! We address your concerns and revise the paper accordingly as follows:
> 1. Contribution: The optimal sample complexity for AMDP is a significant and challenging open problem in RL theory. A number of related publications in notable venues have appeared in the last six years (see Table 1). We offer a new algorithm and a novel analysis that combined yield the provably first min-max optimal sample complexity method for uniformly ergodic MDPs. We strongly believe that our result holds significant theoretical value and is therefore worth publishing. In fact, it is hard to argue the opposite in view of the importance of this problem. We acknowledge the useful ideas introduced in earlier papers. But clearly, key (non-trivial!) additional insights were required to solve this problem; otherwise, a minmax optimal result would have been offered already by one of the previous authors who have worked on this problem. Our contribution requires skillfully combining and extending multiple algorithm ideas and analysis techniques. We highlight these contributions earlier in the paper, in our revised abstract, and in the third paragraph of the introduction.
> 2. Main challenge: A crucial idea that allows us to achieve optimality, compared to the previous approaches, is the realization that the optimal complexity dependence for Discounted MDPs (DMDPs) should be $\widetilde O(t_{mix}(1-\gamma)^{-2}\epsilon^{-2})$, c.f. Wang et al. (2023). However, from an algorithm design standpoint, this is not enough. This needs to be coupled with a sufficiently small initialization sample size. Therefore, the main challenge to our algorithm design and analysis is to achieve a $\widetilde O(t_{mix}(1-\gamma)^{-2}\epsilon^{-2})$ sample complexity with a small minimum sample size. This is explained in the Introduction, Sections 3, and 3.1. These are key insights that are non-trivial to obtain and exploit in algorithmic design, as we explain and illustrate in the numerical results added in the paper.
> 3. Algorithm design: The idea of solving an Average reward MDP (AMDP) by reducing it to a DMDP and applying dynamic programming has been considered and implemented since the 1970s, see Hordijk and Tijms (1975). If this insight was enough, optimal sample complexity would have been achieved in the literature. However, all of the existing papers in RL that use this strategy (Jin and Sidford 2021, Wang et al. 2022) have a $\widetilde O(\epsilon^{-3})$ dependence, which yields a convergence rate of order $O(n^{-1/3})$, a significant deviation from the canonical convergence rate of $O(n^{-1/2})$. Moreover, as shown in the newly added numerical experiment 1, this $\widetilde O(\epsilon^{-3})$ dependence of the prior works is not just an artifact of the analysis, but an issue intrinsic to their algorithm design. We are able to resolve this issue with our new algorithm through more sophisticated analysis techniques.

---

> > ### Comment · Reviewer_fwUc · 2023-11-22
> >
> > The authors addressed most of my problems and I improved my score to 6.

---

### Author Response · Authors · 2023-11-17
**Paper revision updates**

We thank the reviewers for their insightful comments and questions. Your feedback provided valuable suggestions and directions for improvement, leading to the updated version of the paper. We offer a brief summary highlighting the main changes we made, as reflected in the most recently uploaded version.
- We made several changes to the abstract, introduction, and subsequent sections to emphasize the fact that our contribution settles an important open problem in RL sample complexity theory, namely the question of whether there exists an algorithm that achieves the existing average reward lower bound in the literature.
- In Section 4, we added two numerical experiments that separately validate the sample complexity dependence of our proposed algorithm on $\epsilon$ and $t_{minorize}$, yielding Figure 1. As explained in Section 4, both plots align with our sample complexity upper and lower bounds, confirming the theoretical insights in the paper.
- In the first set of experiments, we compared our algorithm with that in Jin and Sidford (2021). We observed a significant empirical improvement in terms of sample efficiency, indicating a distinction between our algorithm and that in Jin and Sidford (2021). This experiment also verifies that our algorithm possesses the canonical convergence rate of $n^{-1/2}$, as compared to the $n^{-1/3}$ rate associated with Jin and Sidford (2021).

---

### Author Response · Authors · 2023-11-21

Thank you very much for your comments and questions. We believe that we have addressed the concerns raised through the updated version of the paper and the follow-up responses. Should you have any further questions or require additional clarification, we would be delighted to further discuss them prior to the discussion period deadline tomorrow, Nov 22nd.

If our updates have addressed and resolved the concerns you initially raised, we kindly request your consideration in adjusting the score accordingly.

---

### Meta-Review · Area_Chair_PzoS · 2023-12-04

**Metareview:**

This paper studies the sample complexity of the ADMP problem under the uniform ergodicity assumption and the generator setting. An algorithm with $O((1-\gamma)^{-2}t_{mix}\epsilon^{-2})$ complexity is designed and analyzed, which matches the existing lower bound of this problem. This not only proves optimality of the proposed algorithm, but also shows the tightness of the existing lower bound. Though the paper has a slight novelty weakness in the algorithm design, as the proposed approach is a mixture of 3 existing works: (Jin & Sidford, 2021), (Li et al. 2020), and (Wang et al. 2023), the paper is still worth publishing as it closes a gap in the complexity theory of AMDP.

**Justification For Why Not Higher Score:**

The paper has a slight novelty weakness in the algorithm design, as the proposed approach is a mixture of 3 existing works: (Jin & Sidford, 2021), (Li et al. 2020), and (Wang et al. 2023).

**Justification For Why Not Lower Score:**

The paper closes a gap in $t_{mix}$ for the complexity theory of AMDP.

---

### Decision · Program_Chairs · 2024-01-16

Accept (poster)